# Use of bio-information and communication technology shortens time to peak at a lower height of the epidemic curve: An alternative to flattening for countries with early COVID-19 outbreaks

**Song Hee Hong**[1,2☯*], **Xinying Jiang**[1¤a], **HyeYoung Kwon**[1☯¤b]

1 Institutional and Regulatory Science in Pharmacy, Seoul National University, Seoul, South Korea,
2 Research Institute of Pharmaceutical Sciences, Seoul National University, Seoul, South Korea

☯ These authors contributed equally to this work.
¤a Current address: Healthcare and Life Sciences in China and Renaissance Group, Shanghai, China
¤b Current address: Department of Public Health, Mokwon University, Daejeon, South Korea
* songhhong@snu.ac.kr

**Data Availability Statement:** All relevant data are within the manuscript.

## Abstract

### Introduction

The traditional approach to epidemic control has been to slow down the rate of infection while building up healthcare capacity, resulting in a flattened epidemic curve. Advancements in bio-information-communication technology (BICT) have enabled the preemptive isolation of infected cases through efficient testing and contact tracing. This study aimed to conceptualize the BICT-enabled epidemic control (BICTEC) and to document its relationships with epidemic curve shaping and epidemic mitigation performance.

### Methods

Daily COVID-19 incidences were collected from outbreak to Aug. 12, 2020, for nine countries reporting the first outbreak on or before Feb. 1, 2020. Key epidemic curve determinants–peak height (PH), time to peak (TTP), and area under the curve (AUC)–were estimated for each country, and their relationships were analyzed to test if epidemic curves peak quickly at a shorter height. CFR (Case Fatality Rate) and CI (Cumulative Incidence) were compared across the countries to identify relationships between epidemic curve shapes and epidemic mitigation performance.

### Results

China and South Korea had the quickest TTPs (40.70 and 45.37 days since outbreak, respectively) and the shortest PHs (2.95 and 4.65 cases per day, respectively). Sweden, known for its laissez-faire approach, had the longest TTP (120.36) and the highest PH (279.74). Quicker TTPs were correlated with shorter PHs ($\rho = 0.896$, $p = 0.0026$) and lower AUCs (0.790, $p = 0.0028$), indicating that epidemic curves do not follow a flattened

**Funding:** The author(s) received no specific funding for this work.

**Competing interests:** NO authors have competing interests.

trajectory. During the study period, countries with quicker TTPs tended to have lower CIs ($\rho$ = .855, P = .006) and CFRs ($\rho$ = 0.684, P = .061). For example, South Korea, with the second-quickest TTP, reported the second lowest CI and the lowest CFR.

## Conclusions

Countries that experienced early COVID-19 outbreaks demonstrated the epidemic curves that quickly peak at a shorter height, indicating a departure from the traditional flattened trajectory. South Korea's BICTEC was found to be at least as effective as most lockdowns in reducing CI and CFR.

## Introduction

Countries grappling with the initial outbreak of SARS-CoV-2 (Severe acute respiratory syndrome coronavirus 2) deployed diverse strategies to minimize the risk of COVID-19 [1–3]. Notably, nations such as the UK and France embraced a mitigation strategy, focusing on reducing fatality rates through medical care for severe cases and implementing social distancing measures to flatten the epidemic curve [4]. In contrast, other countries like South Korea and Germany opted for a containment strategy, aiming to disrupt the chain of transmission through a blend of testing and isolation policies [4, 5]. Particularly in Korea, the containment strategy has evolved into a bio-information-communication technology (BICT)-based contact tracing model, enabling the preemptive isolation of infected cases through efficient testing and exhaustive tracing [6]. These divergent approaches underscore the compelling need for an investigation into their respective impacts on the epidemic curve and case fatality. While many cross-country comparisons have been reported in terms of incidence and case fatality [3, 4, 7, 8], few studies have linked the BICT model to the shape of the epidemic curve and case fatality.

Contact tracing, a crucial component in the control of sexually transmitted diseases (STDs), has not garnered as much attention in epidemic control as other measures like lockdowns and social distancing [9]. The limited role of contact tracing in epidemic control is attributed to the rapid surge of epidemics, making it challenging to identify infection suspects through this method, unlike STDs. For instance, contact tracing would need to cover more than 70% of the contacts to effectively contain an epidemic with a reproduction number of 2.5, which is not feasible with the traditional approach of contact tracing [2].

Modern contact tracing, represented by high-tech solutions, utilizes Information and Communication Technology (ICT) to build a high-speed network of digital footprints recording real-time personal and/or business transactions [10]. Bio-Technology (BT) has contributed to the development of testing methods for prompt and accurate infection screening. In the context of COVID-19, symptom-based screening leaves gaps for the transmission of asymptomatic infections. In comparison, the polymerase chain reaction (PCR) test for COVID-19 provides accurate results within no more than two days, and some newer tests take less than an hour [11]. The BICT-based contact tracing is thus capable of preemptively isolating only those infected cases without necessitating strict social distancing.

Of particular interest is to investigate whether the BICT contact tracing shapes the epidemic curve differently that the traditional flattened one. While numerous studies have explored various shapes of the flattened curve, most of them are variants of the flattened curve that can be explained based on the Susceptible-Infectious-Recovered (SIR) model or its analogs [12–14]. Few studies have proposed an alternative shape that is distinct from the flattened curve. A

recent study published in The Lancet described two different epidemic trajectories: one where peak infection occurs sooner at a shorter height, and another with a flattened normal trajectory [2]. However, the study did not establish a connection between these trajectories and different strategies for epidemic control. The former was simply linked to effective social distancing, while the latter was linked to typical social distancing. Furthermore, the study did not elaborate on what constitutes effective social distancing. Therefore, the objective of this study is to investigate the relationship between epidemic trajectories and epidemic control strategies. The results of this study would inform the design of effective strategies to contain initial surges of cases in epidemic outbreaks in a highly-connected modern society.

The specific aims of this study were to conceptualize BICT-enabled contact tracing and document its empirical evidence by estimating country-specific epidemic curves for the nine countries that reported the first case of COVID-19 on or before February 1, 2020 [2]. The study further aimed to explore the relationship among curve characteristics such as Peak Height (PH), Time to Peak (TTP), and Area Under the Curve (AUC) to elucidate whether epidemic curves peak quickly at a shorter height as opposed to flattening out. Additionally, the study sought to compare Case Fatality Rates (CFR) and Case Incidence (CI) across the countries to identify relationships between epidemic curve shapes and epidemic control performance.

## Methods

### Conceptual framework

The BICT contact tracing model would shape the epidemic curve such that it peaks quickly at a shorter height (PQSH curve) because the infection surge at the initial stage is quickly dampened through the pre-emptive isolation of infection sources [2, 15]. Incomplete identification however would leave some infections lingering on over time. Therefore, subsequent waves may develop unless artificial herd immunity is established via vaccination (Fig 1).

The relationships between AUCs and peak positions are unique to each curve trajectories (Table 1). Assuming the epidemic curves all move on a PQSH trajectory, AUC becomes smaller as TTP occurs sooner. In other words, cumulative incidence is greatly reduced with a hastened TTP under the assumption of PQSH trajectory. In comparison, under the assumption of a flattened trajectory, AUC does not change with peak positions [16]. The correlation between AUC and TTP thus is positive in the PQSH trajectory but is not in the flattened trajectory. As for the relationship between PH and TTP, PH becomes shorter with a decrease in TTP in the PQSH trajectory. However, the relationship is reversed in the flattened trajectory. The correlation between PH and TTP thus is positive in the PQSH trajectory but negative in the flattened trajectory.

Case fatality, a measure of disease severity, becomes worse as active cases overrun healthcare capacity. Under the assumption of the PQSH trajectory, case fatality would be reduced with hastened TTP because surging daily incidences are quickly arrested before they begin constraining the inadequate healthcare capacity at the initial stage [17]. However, under the assumption of the flattened trajectory, case fatality would be reduced with extended TTP because the healthcare capacity that rises over time can effectively accommodates the cases that are dispersed.

Lastly, trajectories of the epidemic curve would predict how long the mitigation strategy should last. The mitigation effort stops when herd immunity is established. Herd immunity is achieved naturally under the flattened curve, but achieved artificially with vaccines under the PQSH curve. While both curves need a vaccine to quickly establish the herd immunity, only

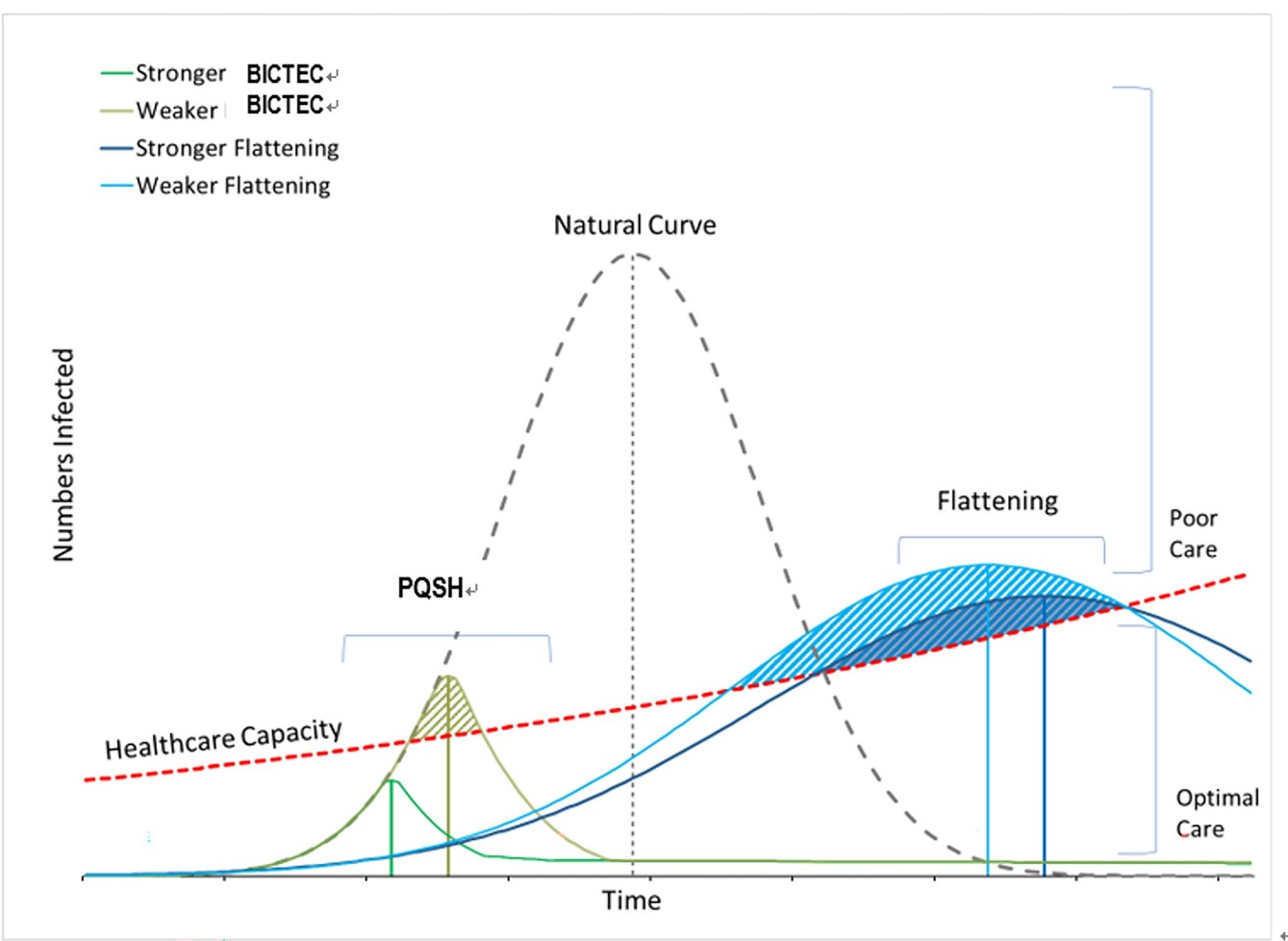

**Fig 1. Conceptual framework for the PQSH vs. flattened trajectories of the epidemic curve.** BICTEC: Bio-Information Communication Technology enabled Epidemic Control, PQSH: Peaks Quickly at a Shorter Height.

**Table 1. Comparison of two models of epidemic control.**

|  | BICTEC | Public Health Model[ξ] |
|---|---|---|
| Epidemic Curve Trajectory | PQSH | Flattened |
| Mitigation Strategy | Testing and Contact Tracing | Social Distancing |
| Curve Determinants |  |  |
| TTP | Quickened | Extended |
| PH | Lowered | Lowered |
| AUC | Reduced | Constant |
| Reduction of Case Fatality | Incidence Containment | Capacity Building |
| Herd Immunity | Vaccine | Natural Infection |

[ξ]: The concept of flattening the curve is described as the public health model [18].

BICTEC: Bio-Information and Communication Technology-enabled Epidemic Control

PQSH: Peaks Quickly at a Shorter Height

TTP: time to peak, PH: peak height, AUC: area under the curve

the PQSH curve predicts a subsequent wave of infections when a vaccine is not introduced in time.

## Measurement of epidemic curve determinants (TTP, PH, and AUC)

A viral infection such as COVID-19 follows a natural life cycle of growth. The total number of COVID-19 cases over time thus can be modelled using a sigmoidal curve. This study used the Gompertz equation below to represent the sigmoidal growth curve of COVID-19 infections.

$$Y(t) = A^* exp(-B^* exp(-C^* t)),$$

where $Y(t)$ is the total number of COVID-19 cases at time $t$, $A$ is the asymptotic maximum number of cases, $B$ denotes how the curve shifts along the time axis, and $C$ represents the growth-rate [19].

A Gompertz curve to an epidemic curve is what cumulative incidence is to daily incidence. Therefore, the peak position of an epidemic curve (Fig 1) is an inflection point on the Gompertz curve. The TTP in the epidemic curve is thus computed as $Ln(B)/C$ from the above parameter estimates [19]. Likewise, the PH of the epidemic curve is computed as $A^*C/e$. For the purpose of international comparison, PH is interpreted as the maximum daily incidence or peak incidence per million population at risk [19].

$AUC(t)$, the area under the epidemic curve from the start of an outbreak to a time $t$, represents the sum of all new cases up to the time $t$. It is simply the $Y(t)$ from the Gompertz equation. Because of the symmetry of the epidemic curve, the AUC at TTP represents half the entire AUC. The AUC taken as the proportion out of the million population at risk indicates a country's cumulative incidence estimated per million people for the purpose of international comparison.

## Cumulative incidence and case fatality rate

CI(t), the cumulative incidence from the start of an outbreak to a time t, represents the proportion of all new cases up to the time t out of the population at risk. Here, the CI at the peak was computed by adding all the new cases reported over TTP per million population.

The CFR measures disease severity and is widely used to evaluate the performance of a country's epidemic control strategies. However, it is inconsistent over time, especially in the case of COVID-19 where healthcare capacity constraints change depending on daily incidences. Due to its inconsistency over time, the CFR at the peak was used for international comparison. It was calculated as total deaths per total cases reported from the start of the outbreak to TTP and expressed as a percentage.

## Selection of countries

The countries covered in this study were selected based on outbreak dates and data availability. How prepared a country is for COVID 19 would differ depending on how long it has been since the outbreak. This study selected the countries that faced the initial outbreak on or before February 01, 2020. These countries include the UK, the US, Germany, France, Spain, Italy, Sweden, China, and South Korea. These nine countries accounted for 32.7% and 43.8% of the global cases and deaths, respectively, as of 12 August 2020 [20].

## Data source

Time series data capturing daily COVID-19 cases and deaths from December 31, 2019, to August 12, 2020, were provided by Our-World-In-Data [21]. These figures represent

confirmed cases and deaths reported originally to the World Health Organization (WHO) and subsequently revised by Our-World-In-Data, with the numbers contingent upon the extent of testing conducted. This dataset is considered rich and comprehensive, frequently cited by prominent news media outlets such as The New York Times and CNN, and commonly utilized in research [22]. For each of the selected countries, a variable of time was created to indicate the number of days from the date of the outbreak ($t = 1$) to the end of the observation. Thus, the observation period varied from 194 days for the UK to 226 days for China. The size of the population at risk in each of the selected countries was also obtained from Our-World-In-Data.

### Analysis

The determination of whether individual countries' epidemic curves follow a flattened trajectory or exhibit a shape resembling PQSH was conducted by analyzing correlations among key curve determinants, including Time to Peak (TTP), Peak Height, and Area Under the Curve (AUC). These determinants were estimated under the assumption of a single sigmoidal growth cycle. The hypothesis of a flattened curve was rejected if these correlations were found to be significantly positive at an alpha level of 0.05.

To assess the relationship between epidemic curve shapes and epidemic control performance (Case Incidence (CI) and Case Fatality Rate (CFR)), a graphical analysis was undertaken. The objective was to investigate whether countries with a quicker Time to Peak (TTP) also reported lower CI and CFR. The CI and CFR values used for this study were those measured at the Time to Peak (TTP). All analyses were performed using SAS Windows (version 9.4)

## Results

### Country-specific epidemic curves

The number of COVID-19 cases that accumulated since the outbreak in each country was modelled using the Gompertz function that represents an S-shape of the growth curve (Fig 2). For most of the countries studied, the Gompertz function showed a sufficient level of goodness of fit. However, poor goodness of fit was seen in the case of the US where the epidemic curve had not reached the peak yet. For South Korea, Spain, and France, the goodness of fit was quite sufficient except for the tail end where a small number of new cases kept occurring beyond the fitted line.

A country's epidemic curve can be best characterized by its peak position (TTP and PH). China which reported the first outbreak had the quickest TTP (40.70 days) followed by South Korea (45.37 days) (Table 2). Sweden, on the other hand, had the longest TTP (120.36 days). Notably, for the US, the TTP has been estimated at 4,357.14 days which means that the peak has yet to be reached.

The PH of an epidemic curve indicates the largest daily count of new cases. Because the daily count depends on the population size, it is necessary to take the count as a proportion of the population to make an international comparison. Thus, the PH or peak incidence was computed per 1 million population. China showed the lowest peak incidence of 2.95 cases per million followed by South Korea's 4.65. The US had the highest peak incidence of 445,202 followed by Sweden (279.74), Spain (117.25), and France (43.76).

### Trajectories of epidemic curves

The correlation between TTP and PH indicates whether epidemic curves have moved on a PQSH or a flattened trajectory. The correlation was strongly positive ($\rho = 0.896$, $p = 0.0026$),

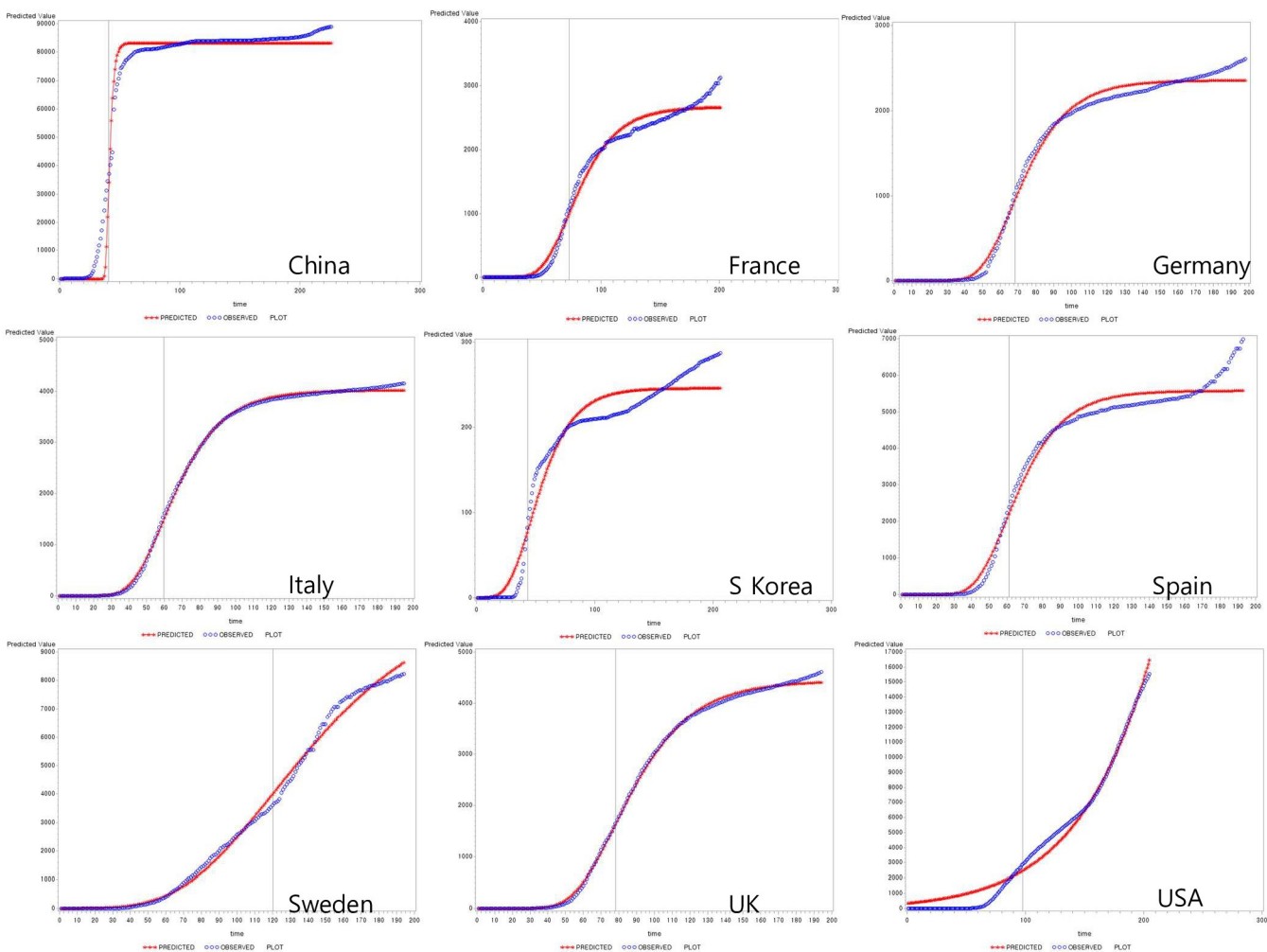

**Fig 2. Total cases and best-fit Gompertz curve.**

which indicates that TTP becomes longer with a taller PH (Table 3), the characteristic of a PQSH trajectory. The two quickest TTPs of 40.70 and 45.37 days since the outbreak were observed for the two lowest peak incidences of 2.95 and 4.65 cases per million respectively (Table 2). Conversely, the longest TTPs of 1,784.39 and 120.36 were for the highest peak incidences of 4,453.02 and 279.74, respectively.

Another way to determine a curve trajectory is to look at the correlation between AUC and TTP. Here as well, the correlation (ρ = 0.864, p = 0.0028) was significantly positive, which supports that the epidemic curves have moved on a PQSH trajectory.

The associations of TTP with PH as well as with AUC are evident (Figs 3 and 4); i.e., not only PH but also AUC clearly increases with an increase in TTP. Sweden with the longest TTP had the largest AUC. China with the shortest TTP had the smallest AUC (25.9). South Korea with the next shortest TTP has the next smallest AUC (103.9).

## International comparison of CFR and CI

Sweden with the longest TTP had the second highest CFR after UK. South Korea having the second quickest TTP next to China had the lowest CFR. However, China, despite its quickest

**Table 2. Estimation of epidemic curve determinants for individual countries.**

| Country | Outbreak (Date) | Trajectory Determinants | Estimate | Degree Freedom | Lower | Upper |
|---|---|---|---|---|---|---|
| China | (31 Dec.) | TTP | 38.52 | 226 | 38.25 | 38.79 |
| | | Height | 2.95 | 226 | 2.81 | 3.08 |
| | | AUC | 43.08 | 226 | 42.92 | 43.23 |
| Korea | (20 Jan.) | TTP | 45.97 | 206 | 44.51 | 47.43 |
| | | Height | 4.65 | 206 | 4.02 | 5.27 |
| | | AUC | 180.92 | 206 | 177.23 | 184.62 |
| Italy | (31 Jan.) | TTP | 59.50 | 195 | 59.29 | 59.71 |
| | | Height | 81.70 | 195 | 80.50 | 82.90 |
| | | AUC | 2957.49 | 195 | 2949.18 | 2965.79 |
| Spain | (1 Feb. | TTP | 59.72 | 193 | 58.70 | 60.74 |
| | | Height | 117.25 | 193 | 107.02 | 127.47 |
| | | AUC | 4104.18 | 193 | 4041.56 | 4166.81 |
| Germany | (28 Jan.) | TTP | 66.46 | 198 | 65.83 | 67.08 |
| | | Height | 49.10 | 198 | 46.64 | 51.56 |
| | | AUC | 1733.82 | 198 | 1717.87 | 1749.78 |
| France | (25 Jan.) | TTP | 72.59 | 201 | 71.56 | 73.62 |
| | | Height | 43.76 | 201 | 40.82 | 46.70 |
| | | AUC | 1964.56 | 201 | 1933.18 | 1995.95 |
| UK | (31 Jan.) | TTP | 78.07 | 194 | 77.79 | 78.35 |
| | | Height | 69.32 | 194 | 68.28 | 70.37 |
| | | AUC | 3265.11 | 194 | 3250.86 | 3279.35 |
| Sweden | (1 Feb.) | TTP | 112.14 | 194 | 108.58 | 115.71 |
| | | Height | 279.74 | 194 | 270.09 | 289.38 |
| | | AUC | 5137.75 | 194 | 4911.11 | 5364.38 |
| US | (21 Jan.) | TTP | 1965.59 | 205 | 1921.07 | 2010.11 |
| | | Height | 1029075 | 205 | 1003472 | 1054678 |
| | | AUC | 147280000 | 205 | | |

Abbreviation: TTP: time to peak, Height: Incidence per million population at Peak

*p < .0001.

TTP had a relatively higher CFR. Further, Germany with a relatively longer TTP had a relatively lower CFR.

## Discussion

The main finding of this study is that the nine countries' epidemic curves exhibit the characteristics of the PQSH trajectory. The correlation between the TTP and PH ($\rho = 0.896$, p = 0.0026) as well as the one between the PH and AUC ($\rho = 0.894$, p = 0.0028) was strongly positive,

**Table 3. Correlations between TTP, PH, and AUC.**

| | TTP | PH | AUC |
|---|---|---|---|
| **TTP** | 1 | 0.896* | 0.864* |
| **PH** | | 1 | 0.957* |
| **AUC** | | | 1 |

*p < 0.01. TTP: time to peak, PH: peak height, AUC: area under the curve.

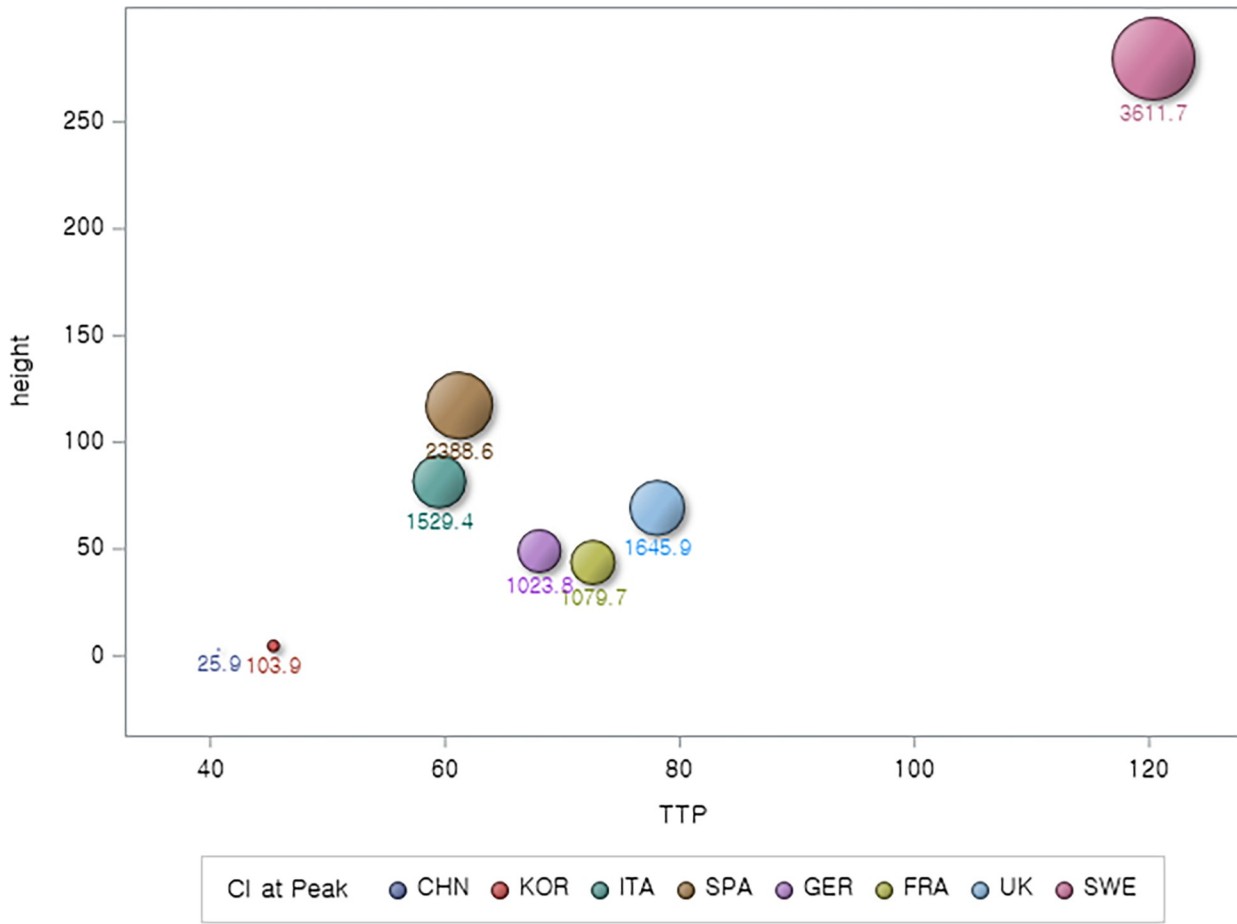

**Fig 3. Mapping countries on epidemic curves' peak positions (TTP and PH) with bubble representing CI up to the peak.**

supporting for the PQSH trajectory hypothesis over the flattened curve. An epidemic curve, if flattened, should have displayed a longer TTP than the natural curve closely approximated by Sweden's laissez-faire approach [23, 24]. While a direct comparison with Sweden may not be possible due to different country-specific conditions, no country except for the US had a TTP longer than Sweden. This implies that the countries experiencing the early outbreak of COVID 19 had prioritized a strategy that rapidly quickens the peak rather than prolonging it.

The widely accepted goal of epidemic control is to decelerate and elongate the spread of infections over an extended period, allowing for capacity building—a concept embodied in the flattened curve [25–27]. However, when examining graphical representations of new daily cases across various countries, there is limited evidence supporting the existence of a flattened curve [28]. In fact, all curves exhibit multiple waves with relatively tall peaks. Each wave, including the initial one, lacks the characteristics of a flattened curve, which, if present, should have a width wider than its height.

Many studies have challenged the flattened curve theory, scrutinizing aspects such as the vertical axis label, the number of peaks, curve shapes, and the overall duration of the pandemic [29–32]. Notably, a study reports that the curve is not merely flattened but actually shrunk [31]. Another study, published in The Lancet, has introduced an alternative model of epidemic transmission [2], which aligns with our model of BICT-enabled epidemic control.

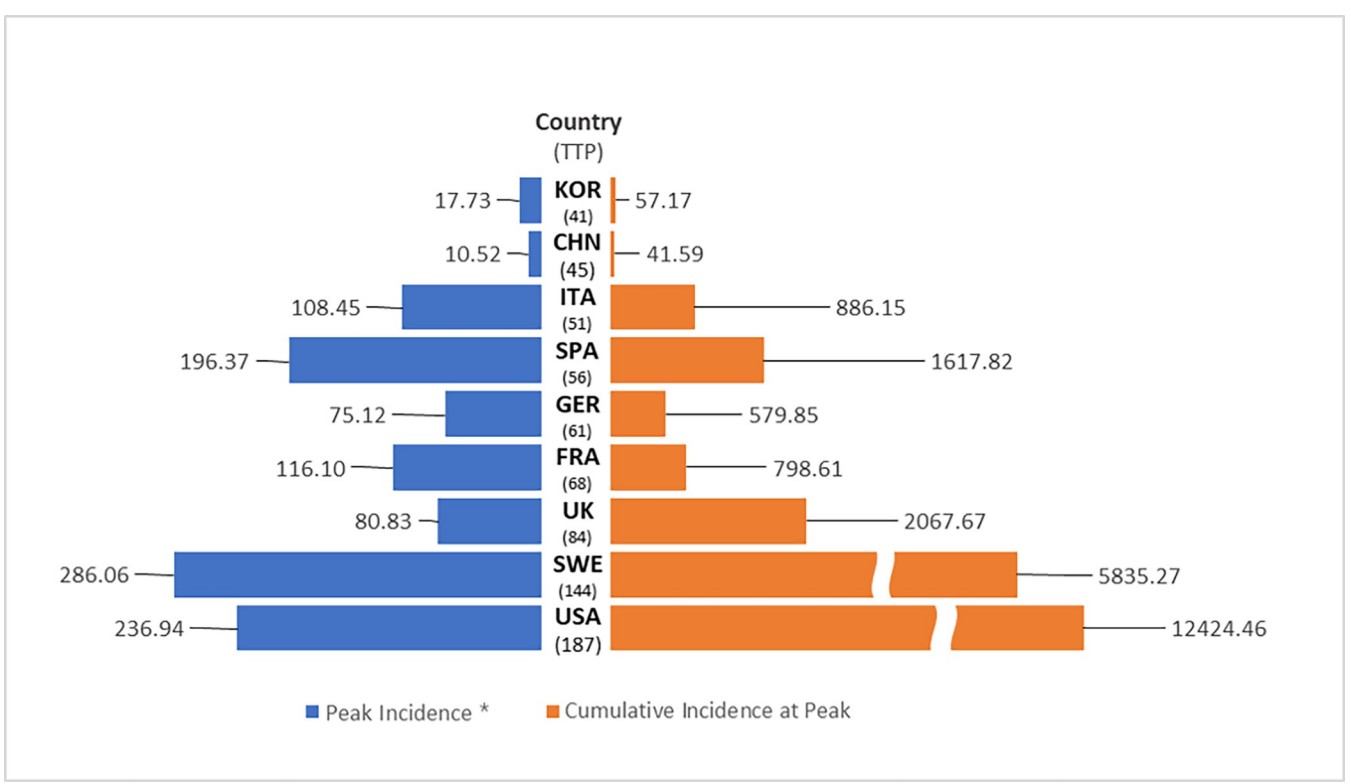

**Fig 4. Comparison of CFR and CI across countries.** Correlation between TTP and CFR = .684 (P = .061) and correlation between TTP and CI = 0.855 (P = .006).

In addition to the strong correlation observed between faster TTPs and shorter PHs, the further identification of a strongly positive correlation between faster TTPs and lower AUCs lends support to the PQSH trajectory. Under the flattened trajectory, AUCs remain consistent regardless of TTPs. However, under the PQSH trajectory, not only does the PH decrease, but the AUC also diminishes as the peak infection arrives sooner.

The intriguing question emerges as to whether faster TTPs leads to lower fatalities. According to our findings, faster TTPs were associated with lower CIs, measured as the sum of daily new cases up to the TTP ($\rho$ = .855, P = .006). However, the correlation between TTP and CFR was not as robust ($\rho$ = 0.684, P = .061). Perhaps, CFR may depend not only on how quickly surging infections are dampened but also on how well public health is prepared for the infection. South Korea, with the second-fastest TTP, had the lowest CFR, outperforming Germany —a country known for its public health preparedness [4]. In contrast, China, with the fastest TTP, had a higher CFR than Germany, despite implementing a complete lockdown for a zero-tolerance policy. This indicates that the speed of halting infections is not the sole determinant of CFR; rather, it might be influenced by the effectiveness of public health preparedness measures.

Although China and South Korea exhibited similar curve characteristics in terms of TTP, Peak Height (PH), and AUC, the two countries employed different strategies. China implemented a city-wide lockdown, while South Korea emphasized test-based triaging [1, 33, 34]. The complete lockdown quickly shortened TTP by isolating all residents without tracing and testing infection suspects; complete lockdown can be viewed as a special case of our model where all the people in a boundary are assumed to be cases and therefore isolated. However,

the lockdown disabled social and business functions in the community. In contrast, test-based triaging assumes that individuals suspected of infection can be individually identified and targeted for testing and triaging. This method quickened TTP almost as rapidly as the complete lockdown in China while allowing many social and business functions to continue. Moreover, South Korea's TTP was quicker than all European countries that implemented a lockdown, although it was not as restrictive as China's.

These findings suggest a paradigm shift in epidemic control strategies, diverging from traditional curve-flattening approaches. Instead of merely slowing down transmission, the emphasis should be on halting it as swiftly as possible to minimize infections and fatalities. In our interconnected world, news of casualties, even in a remote area, spreads rapidly and becomes a personal concern for neighbors. This urgency prompts governments to take swift action to resolve the situation as quickly as possible. The crucial questions arising are how to achieve this.

The BICTEC system executed in South Korea began with a stockpile of adequate COVID-19 testing capability [35, 36]. The system then aimed to pre-empt the community spread of COVID-19 by tracing all the contacts of each infected person as soon as possible. The contacts were identified using digital footprints, and were notified via text messages to get tested. Those who tested positive were triaged for isolation, hospital care, or intensive care services, depending on the severity of COVID-19 [37]. The ICT-based contact tracing despite concerns on privacy violation gained consensus among constituencies for a greater benefit of preventing infection transmission [38]. All tests and treatments were paid from national health insurance programs.

Technological advancements have undoubtedly contributed to the successful execution of the BICT-based epidemic control model. In the past, the identification of infected persons was based on symptoms rather than on bio-tech testing. Prompt tracing of all infection sources and their links requires the use of ICT to identify those who are exposed and to inform them to get tested. The success of BICTEC also requires the timely introduction of a safe and effective vaccine, which was made possible from advances in BT. Without the introduction of the vaccine, BICTEC would leave many people vulnerable to subsequent waves of infections.

This study has certain limitations that should be acknowledged. Firstly, given the ongoing nature of the COVID-19 epidemic, the number of confirmed cases is subject to change, possibly involving multiple waves. Consequently, the findings of this study are particularly relevant to the initial responses to epidemic outbreaks.

Secondly, the reported number of cases might be under-estimated due to cases not being identified as suspects for testing. The extent of under-reporting could vary across countries, influenced by factors such as testing strategies and capacities. For instance, South Korea implemented widespread testing with a capacity of 20,000 tests per day [35], while the UK relied on symptom-based testing [5, 39, 40].

Thirdly, the trajectory of the epidemic is likely influenced by various factors beyond the epidemic control models investigated. Demographics, population density, social practices, and health beliefs, which vary among countries, could impact the transmission dynamics and epidemic trajectory. These factors might also affect case fatality rates, along with healthcare capacities and public health preparedness for the pandemic.

Despite these limitations, the estimated epidemic trajectory reflects each country's unique efforts to control the epidemic.

## Conclusions

The concept of BICTEC, popularized in Korea for preemptive isolation of infected cases through efficient testing and contact tracing, predicts that the epidemic curve will peak quickly

at a shorter height instead of flattening, thereby altering the traditional perspective on epidemic mitigation strategies. Furthermore, the BICTEC is at least as effective as more restrictive lockdowns in reducing CI and CFR.

## Acknowledgments

All the authors would like to express their gratitude toward Ms. Gu Seul for extending her support to this study.

## Author Contributions

**Conceptualization:** Song Hee Hong, HyeYoung Kwon.

**Data curation:** Song Hee Hong, Xinying Jiang.

**Formal analysis:** Xinying Jiang, HyeYoung Kwon.

**Investigation:** Song Hee Hong, Xinying Jiang.

**Methodology:** Song Hee Hong, Xinying Jiang.

**Project administration:** Song Hee Hong.

**Resources:** Xinying Jiang.

**Supervision:** Song Hee Hong.

**Validation:** Song Hee Hong.

**Writing – original draft:** Song Hee Hong, HyeYoung Kwon.

**Writing – review & editing:** Song Hee Hong, Xinying Jiang.

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
