## [Decision Letter · Decision Letter 0]

18 Oct 2023

PONE-D-23-10043Use of Bio-Information and Communication Technology Shortens Time to Peak at a Lower Height of the Epidemic Curve: An Alternative to Flattening for Countries with Early COVID-19 OutbreaksPLOS ONE

Dear Dr. Hong,

Thank you for submitting your manuscript to PLOS ONE. After careful consideration, we feel that it has merit but does not fully meet PLOS ONE’s publication criteria as it currently stands. Therefore, we invite you to submit a revised version of the manuscript that addresses the points raised during the review process. Please improve your manuscript based on the comments of the two reviewers. Reviewer 2 also recommended the inclusion of two references. According to the policies of PLOS ONE, you are not obliged to include these, unless you feel that they are highly relevant to your manuscript. Please submit your revised manuscript by Nov 25 2023 11:59PM. If you will need more time than this to complete your revisions, please reply to this message or contact the journal office at plosone@plos.org. Please include the following items when submitting your revised manuscript:A rebuttal letter that responds to each point raised by the academic editor and reviewer(s). You should upload this letter as a separate file labeled 'Response to Reviewers'.A marked-up copy of your manuscript that highlights changes made to the original version. You should upload this as a separate file labeled 'Revised Manuscript with Track Changes'.An unmarked version of your revised paper without tracked changes. You should upload this as a separate file labeled 'Manuscript'.

We look forward to receiving your revised manuscript.

Kind regards,

Siew Ann Cheong, Ph.D.

Academic Editor

PLOS ONE

Journal Requirements:

Reviewers' comments:

Reviewer's Responses to Questions

**Comments to the Author**

1. Is the manuscript technically sound, and do the data support the conclusions?

Reviewer #1: Partly

Reviewer #2: Yes

2. Has the statistical analysis been performed appropriately and rigorously? 

Reviewer #1: Yes

Reviewer #2: Yes

3. Have the authors made all data underlying the findings in their manuscript fully available?

Reviewer #1: Yes

Reviewer #2: Yes

4. Is the manuscript presented in an intelligible fashion and written in standard English?

Reviewer #1: Yes

Reviewer #2: Yes

5. Review Comments to the Author

Reviewer #1: The manuscript aim is to conceptualize the BICT-enabled epidemic control (BICTEC) and to document its relationships with epidemic curve shaping and epidemic mitigation performance.

The quality of the manuscript need to be improved by incorporating the following comments,

- Motivation, State of the art & Research questions - related to objective of the work need to be presented in Introduction Section

- Related works need to be presented and compared.

- Justify how the input data considered is sufficient for proposed work.

- Justify how the input data is sufficient to address the proposed work objective.

- Justify how the considered input and analysis performed matches to the proposed work objective.

- How the presented result analysis is considered as correct/efficient compared exisiting analysis in related works?

Overall, The input considered, analysis performed & results presented is not sufficient to justify BICT-enabled epidemic control is efficient.

Reviewer #2: Thank you for submitting this very interesting work. However, I would like to point out some improvements to be made.

1. More quantitative information should be provided in the abstract.

2. Please provide the keywords after the abstract and write them alphabetically.

3. In the introduction, please provide explanations about the COVID-19 epidemic and its control strategies in the world. For this, you can use the following studies:

- Environmental Health Engineering and Management Journal. 2022 Jan 10;9(1):41-53. http://ehemj.com/article-1-901-en.html

- Toxicology and Industrial Health. 2021 Jun;37(6):353-64. https://doi.org/10.1177/07482337211013319

4. The discussion is brief. Please strengthen this section and compare the results of your study with studies conducted in other countries.

5. The conclusion is very brief. Please indicate what implications the findings of this study hold for future research directions, policy-making, or public health strategies?

6. PLOS authors have the option to publish the peer review history of their article (what does this mean?). If published, this will include your full peer review and any attached files.

Reviewer #1: No

Reviewer #2: **Yes: **Mahdiyeh Mohammadzadeh

---

## [Author Response · Author response to Decision Letter 0]

17 Jan 2024

Reviewer #1: The manuscript aims to conceptualize the BICT-enabled epidemic control (BICTEC) and document its relationships with epidemic curve shaping and epidemic mitigation performance. The quality of the manuscript needs improvement by incorporating the following comments.

1. Motivation, State of the art & Research questions - related to the objective of the work need to be presented in the Introduction Section

• Response: Thank you for pointing out these weaknesses. We've addressed this by adding a paragraph in the introduction section related to the study motivation and research questions.

2. Related works need to be presented and compared.

• Response: Related works are added in the introduction and are compared with our study findings in the discussion section.

3. Justify how the input data considered is sufficient for the proposed work.

• Response: Thank you for this comment. We recognize that the description of the input data is incomplete. We have, therefore, added several sentences in the Data Source subsection to sufficiently describe what the input data are. We have also added some citations to support the input data used.

4. Justify how the input data are sufficient to address the proposed work objective.

• Response: We are not sure whether we have correctly understood this comment, which is given separately from the previous comment. The study objective was to determine whether the epidemic curve follows a flattened trajectory. For this objective, we have estimated key determinants of the epidemic curve and quantified their correlations. This justification is given under the conceptual framework subjection as well as under the analysis subsection which we have added per your comments. The objective related to curve shaping and epidemic mitigation performance is also described in the conceptual framework and analysis subsections.

5. Justify how the considered input and analysis performed match the proposed work objective.

• Response: We have recognized the shortcomings in describing the relationship among the input, the analysis, and the study objective. Thus, we have added a subsection of analysis to link those elements.

6. How the presented result analysis is considered correct/efficient compared to existing analysis in related works?

• Response: Again, we have recognized this weakness in comparing with existing works. Therefore, we have thoroughly revised the discussion section adding the previous work. We have also added a subsection as to whether the BICTEC performed better.

Overall, the input considered, analysis performed & results presented are not sufficient to justify BICT-enabled epidemic control is efficient.

• Response: Thank you for your valuable comments and reviews. Accordingly, we have revised the introduction, the methodology, and the discussion sections. Despite some limitations, we believe that our study findings on the BICTEC contribute to the literature.

Reviewer #2: Thank you for submitting this very interesting work. However, I would like to point out some improvements to be made.

1. More quantitative information should be provided in the abstract.

• Response: Per your comments, we have added quantitative statistics on curve shaping and performance measures such as CFR and CI.

2. Please provide the keywords after the abstract and write them alphabetically.

• Response: We've added five keywords after the abstract.

In the introduction, please provide explanations about the COVID-19 epidemic and its control strategies worldwide. For this, you can use the following studies: 

*Environmental Health Engineering and Management Journal. 2022 Jan 10;9(1):41-53. http://ehemj.com/article-1-901-en.html

*Toxicology and Industrial Health. 2021 Jun;37(6):353-64. https://doi.org/10.1177/07482337211013319

• Response: Thank you for this comment. We've added a paragraph related to study motivation and existing work in the introduction section. Regarding the articles listed below that you have suggested we cite in our paper, they investigated strategies to identify factors affecting the management of the COVID pandemic in businesses such as hospitals and steel complexes. We apologize that we couldn’t cite your articles as a reference since these articles are not quite relevant to our topic. We hope this is acceptable.

3. The discussion is brief. Please strengthen this section and compare the results of your study with studies conducted in other countries.

• Response: We appreciate your comments. We've added several paragraphs to compare our study results with previous work.

4. The conclusion is very brief. Please indicate what implications the findings of this study hold for future research directions, policy-making, or public health strategies?

• Response: We appreciate your comments. We've added a paragraph to describe the implications of our study. Please also note that we have thoroughly revised the discussion section.

---

## [Decision Letter · Decision Letter 1]

16 Feb 2024

PONE-D-23-10043R1Use of Bio-Information and Communication Technology Shortens Time to Peak at a Lower Height of the Epidemic Curve: An Alternative to Flattening for Countries with Early COVID-19 OutbreaksPLOS ONE

Dear Dr. Hong,

Thank you for submitting your manuscript to PLOS ONE. After careful consideration, we feel that it has merit but does not fully meet PLOS ONE’s publication criteria as it currently stands. Therefore, we invite you to submit a revised version of the manuscript that addresses the points raised during the review process.

Specifically, please address the last few comments by Reviewer 1.

We look forward to receiving your revised manuscript.

Kind regards,

Siew Ann Cheong, Ph.D.

Academic Editor

PLOS ONE

Journal Requirements:

Reviewers' comments:

Reviewer's Responses to Questions

**Comments to the Author**

1. If the authors have adequately addressed your comments raised in a previous round of review and you feel that this manuscript is now acceptable for publication, you may indicate that here to bypass the “Comments to the Author” section, enter your conflict of interest statement in the “Confidential to Editor” section, and submit your "Accept" recommendation.

Reviewer #1: All comments have been addressed

Reviewer #2: (No Response)

2. Is the manuscript technically sound, and do the data support the conclusions?

Reviewer #1: Partly

Reviewer #2: Yes

3. Has the statistical analysis been performed appropriately and rigorously? 

Reviewer #1: Yes

Reviewer #2: Yes

4. Have the authors made all data underlying the findings in their manuscript fully available?

Reviewer #1: Yes

Reviewer #2: Yes

5. Is the manuscript presented in an intelligible fashion and written in standard English?

Reviewer #1: Yes

Reviewer #2: Yes

6. Review Comments to the Author

Reviewer #1: The quality of the manuscript need to be improved by addresing the following comments,

- Research Problem, State of the art works, & Research Questions need to be provided in Introduction section

- Related works need to be discussed. Atleast 12-15 works need to be discussed interms of Problem, solution recommended, results obtained and research gap.

- Summary of the related works and scope for improvements need to be provided.

- Novelty of the proposed methodology need to be justified.

- Comparison with exisiting work/methodology need to be provided for justifying the benefits of proposed work/methodology.

Reviewer #2: The requested corrections have been carried out efficiently by the authors and the article is now appropriate for publishing.

7. PLOS authors have the option to publish the peer review history of their article (what does this mean?). If published, this will include your full peer review and any attached files.

Reviewer #1: No

Reviewer #2: **Yes: **M.M.

---

## [Author Response · Author response to Decision Letter 1]

14 Mar 2024

Responses to the reviewer’s comments.

Thank you for your feedback. Our responses have been included below each comment for your convenience. If you have any further concerns or questions, please do not hesitate to let us know. 

Again, thank you for your valuable comments.

Reviewer #1: The quality of the manuscript needs to be improved by addressing the following comments,

- Research Problem, State of the art works, & Research Questions need to be provided in Introduction section

Authors Responses: Thank you for the comments. To elucidate these important concepts, we have revised the fourth paragraph of the introduction section as below. Additionally, we have included references related to the Susceptible-Infectious-Recovered (SIR) model to provide a contrast with our study. Please also note that our research hypotheses are already presented in the methodology under the subheading of analysis.

“Of particular interest is to investigate whether the BICT contact tracing shapes the epidemic curve differently that the traditional flattened one. While numerous studies have explored various shapes of epidemic curves, most of them are variants of the flattened curve that can be explained based on the Susceptible-Infectious-Recovered (SIR) model or its analogs [12-14]. Few studies have proposed an alternative shape that is distinct from the flattened curve. A recent study published in The Lancet described two different epidemic trajectories: one where peak infection occurs sooner at a shorter height, and another with a flattened normal trajectory [2]. However, the study did not establish a connection between these trajectories and different strategies for epidemic control. The former was simply linked to effective social distancing, while the latter was linked to typical social distancing. Furthermore, the study did not elaborate on what constitutes effective social distancing. Therefore, the objective of this study is to investigate the relationship between epidemic trajectories and epidemic control strategies. The results of this study would inform the design of effective strategies to contain initial surges of cases in epidemic outbreaks in a highly-connected modern society.”

- Related works need to be discussed. At least 12-15 works need to be discussed in terms of Problem, solution recommended, results obtained and research gap.

Authors Responses: In our discussion, we made sure to cite related works at least 15 times. Our primary focus was to highlight our findings and provide support for them. Thus, we structured the discussion in a way that our results would support our hypotheses: 1) country-specific epidemic curves follow an alternative shape compared to the flattened one; 2) epidemic control performance (CI and CFR) is associated with this shape. Throughout this process, we incorporated literature evidence to support our results.

To be frank, we found it challenging to fully understand the reviewer's comment regarding the recommended discussion of 12-15 works in terms of problem, solution recommended, results obtained, and research gap. We are not aware of any specific journal guidelines on this recommendation. However, we believe that our discussion adequately addresses the problem (the inadequacy of the flattened curve in reflecting the outcomes of modern epidemic control strategies), the solution recommended (the BICTEC shapes the epidemic curve to peak sooner at a shorter height), the results obtained (our study findings), and the research gap (how our study fills a gap in the literature as well as our study limitations). 

- Summary of the related works and scope for improvements need to be provided.

Authors Responses: We found it challenging to incorporate your recommendation regarding "the provision of a summary of the related works and scope for improvements." We are not aware of any journal guidelines that require the provision of a summary of related works. Our paper did not aim to provide such a summary. Instead, our focus was on presenting empirical evidence for an alternative epidemic trajectory. We therefore highlighted existing research on epidemic control strategies and their limitations in the introduction. Additionally, in the discussion section, we compared our study findings with previous research to contextualize our results. We used statements to maintain a logical flow throughout the paper. We are concerned that providing a summary of the related works may lengthen our research paper unnecessarily and disrupt its logical flow.

As for the scope for improvements, we acknowledge that there is always room for further research and advancements in this field. We have therefore already mentioned the limitations along with potential directions for future research. We hope this clarifies our approach to introducing related works and addressing areas for improvement.

- Novelty of the proposed methodology need to be justified.

Authors Responses: While novelty exists concerning the concept of BICT-based contact tracing, there is no novelty associated with the methodology. In fact, the methodology used to estimate the epidemic curve characteristics is well-documented

- Comparison with existing work/methodology need to be provided for justifying the benefits of proposed work/methodology.

Authors Responses: As for the comparison with existing work, it has been addressed in the discussion section. Regarding the comparison with existing methodologies, it falls outside the scope of our study as outlined in our specific aims. However, the methodology employed to estimate the epidemic curve is thoroughly described in the methodology section, along with relevant citations.

---

## [Editor Report · Decision Letter 2]

20 Mar 2024

Use of Bio-Information and Communication Technology Shortens Time to Peak at a Lower Height of the Epidemic Curve: An Alternative to Flattening for Countries with Early COVID-19 Outbreaks

PONE-D-23-10043R2

Dear Dr. Hong,

We’re pleased to inform you that your manuscript has been judged scientifically suitable for publication and will be formally accepted for publication once it meets all outstanding technical requirements.

Kind regards,

Siew Ann Cheong, Ph.D.

Academic Editor

PLOS ONE
---

## [Editor Report · Acceptance letter]

8 Apr 2024

PONE-D-23-10043R2 

PLOS ONE

Dear Dr. Hong, 

I'm pleased to inform you that your manuscript has been deemed suitable for publication in PLOS ONE. Congratulations! Your manuscript is now being handed over to our production team.

Kind regards, 

on behalf of

Dr. Siew Ann Cheong 

Academic Editor

PLOS ONE